# A Competency Framework to Assess and Activate Education for Sustainable Development: Addressing the UN Sustainable Development Goals 4.7 Challenge

**Naresh Giangrande [1,\*], Rehema M. White [2], May East [3], Ross Jackson [1], Tim Clarke [4], Michel Saloff Coste [5] and Gil Penha-Lopes [6]**

[1]  Gaia Education, Edinburgh EH9 1PL, UK; rossjackson@gaia.org
[2]  School of Geography and Sustainable Development, University of St. Andrews, St. Andrews KY16 9AJ, UK; rmw11@st-andrews.ac.uk
[3]  Town and Regional Planning, University of Dundee, Dundee DD1 4HN, UK; may.east@gaiaeducation.org
[4]  European Network for Community-Led Initiatives on Climate Change and Sustainability (ECOLIS), 1050 Brussels, Belgium; timclarke01@gmail.com
[5]  Institut International de Prospective sur les Ecosystèmes Innovants, Université Catholique de Lille, 59800 Lille, France; msaloff@gmail.com
[6]  Centre for Ecology, Evolution and Environmental Changes, Faculdade de Ciências da Universidade de Lisboa, 1649-004 Lisboa, Portugal; gppenha-lopes@fc.ul.pt
\*  Correspondence: georgegiangrande@gmail.com

**Abstract:** The UN Transforming our World: The 2030 Agenda for Sustainable Development (herein, Agenda 30) and the Sustainable Development Goals (SDGs) offer both a set of aspirations for the kind of future we would like to see for the world and a suite of targets and indicators to support goal implementation. Goal 4 promotes quality education and Target 4.7 specifically addresses Education for Sustainability. However, creating a monitoring and evaluation framework for Target 4.7 has been challenging. The aim of this research was to develop a meaningful assessment process. We used a dialogical intervention across complementary expertises and piloted concepts in a trainer workshop. We then developed a modified competency framework, drawing on previous competency models but innovating through the addition of intrapersonal competencies, a self-reflective validation scheme, a focus on non-formal learning, and specific alignment with SDG 4.7 requirements. Through exploration of how such learning could be activated, we proposed the use of multiple intelligences. Education plays a synergistic role in achieving the aspirations embedded within Agenda 2030 and the SDGs. We concluded that Education for Sustainable Development (ESD) will require individuals to acquire 'key competencies', aligning with notions of transformational learning, in addition to other generic and context specific competencies.

**Keywords:** education for sustainable development; key competencies; sustainable development goals; SDG 4; transformative learning; evaluation framework

## 1. Introduction

Sustainable development can be seen as a process, a means of envisaging and pursuing an aspiration of the future [1]. However, practically agreeing on a process and working towards sustainable development presents multiple challenges. The development and implementation of international policy enables us to collectively decide and plan changes at a societal scale, acknowledging global issues such as loss of biodiversity and climate change. The UN Sustainable Development Goals (SDGs) were launched in 2015, subsequent to the Millennium Development Goals [2]. The SDG process

acknowledged the needs for a more consultative process, recognition of the interconnectedness of sustainability challenges and universality of the SDGs, which apply to all UN member states, as well as across public, private and third sectors [3]. Whilst the SDGs represent a suite of collectively agreed intentions, there are tensions and contradictions within them and wider concerns regarding their framing [4]. However, despite their flaws, they are the best outcome of the political discourse at this time and they offer us a framework to begin to create an 'ecological civilisation'. Together with the SDGs, the UN 2030 Agenda for Sustainable Development offers both a set of aspirations for the kind of future we would like to see and a suite of 17 targets and 169 indicators to support the implementation of these Goals. In this paper, we explore how a competencies framework for Education for Sustainable Development (ESD) supports measurement for this indicator. We propose a greater emphasis on intrapersonal competencies, and we argue for greater consideration of non-formal learning to achieve sustainable development within the context of the SDGs and beyond.

Education is both a Goal in its own right and a means by which other aspects of sustainable development can be achieved. Goal 4 promotes quality education and aims to "ensure inclusive and equitable quality education and promote lifelong learning opportunities for all" [2]. This SDG is concerned with enhancing access to education and equality of access, and to ensuring that there is quality education at all levels to deliver the knowledge and skills for a sustainable future. In this paper we are concerned primarily with SDG 4.7 which aims to, "by 2030, ensure that all learners acquire the knowledge and skills needed to promote sustainable development, including, among others, through education for sustainable development and sustainable lifestyles, human rights, gender equality, promotion of a culture of peace and non-violence, global citizenship and appreciation of cultural diversity and of culture's contribution to sustainable development" [2]. While some commentators consider that "all education is environmental education" [5], there is a shift from emphasising teaching to a focus on learning [6]. Whilst all targets have some indicators assigned, SDG 4.7 in the initial UN documents focused primarily on the following indicator:

> "4.7.1 Extent to which (i) global citizenship education and (ii) education for sustainable development, including gender equality and human rights, are mainstreamed at all levels in: (a) national education policies, (b) curricula, (c) teacher education and (d) student assessment". [2]

Hence, whilst the goal itself acknowledges the need to enable sustainable lifestyles and promotion of a culture of peace, assessment focus is aimed primarily at formal education. This paper acknowledges the contribution of Global Citizenship Education stream of value-based education in the fulfilment of SDG 4.7, but it will only focus on the ESD dimension. Terminology regarding what comprises ESD has been long debated [7,8] and in this paper we employ the UNESCO definition:

> "ESD empowers learners to take informed decisions and responsible actions for environmental integrity, economic viability and a just society, for present and future generations, while respecting cultural diversity. It is about lifelong learning, and is an integral part of quality education. ESD is holistic and transformational education which addresses learning content and outcomes, pedagogy and the learning environment. It achieves its purpose by transforming society." [8]

ESD is considered to be a goal for education rather than a prescribed body of information and knowledge [9]. It takes places in formal, non-formal and informal settings and (as SDG 4.7 suggests) requires it to be integrated into policy, institutional governance, educator training, curriculum and pedagogy and monitoring and assessment [7]. Education involves acquisition of knowledge, skills and capacities. Audit measures do not always capture the quality and depth of education in its capacity to permit socialisation, promoting citizenship in order to support existing society and culture; to be vocational, preparing learners for employment; to achieve liberal goals in enabling individuals to achieve their full potential; and to be transformative, encouraging individual and societal change

towards a better and a fairer world [10]. As well as enabling these ambitious aspirations, assessment methods need to have relevance, validity and feasibility [11].

Educators and researchers need a robust reflective process with regards to ESD in general and Goal 4.7 in particular. We need to question, learn and include this reflection in further developing the intentions and facilitation of ESD. Reflection and learning are required as the pursuit of sustainable development is dynamic, and adaptive learning processes and actions are required. Effective assessment frameworks do not merely add on measurement at the end of a project. Rather, they require deep understanding of the purpose and practice of the goals of an intervention or activity and need to be embedded in the learning cycle of planning, implementation, reflection and re-planning. This approach is particularly important for ESD because sustainability issues are often 'wicked problems': Complex, multifaceted, without one 'correct' answer, and involving many actors [12]. In addition, the normativity of sustainable development creates an intention to move beyond information transfer to knowledge sharing, consideration of values and acquisition of the abilities required to critically reflect on and enact visions for the future [13]. Instead of focusing on the information, or even on skills acquisition, a more pragmatic yet meaningful strategy is to focus on the attributes, capacities or competencies acquired by learners. Here, we thus focus on the concept of competencies to monitor and assess teaching and learning outcomes in relation to SDG 4.7.

Competencies have been understood in different ways. Wiek et al. [13] proposed a synthesis definition of competence as "a functionally linked complex of knowledge, skills, and attitudes that enable successful task performance and problem solving". In this paper we included non-cognitive personal aspects of education and so we also drew on the definition that "a competence is defined as the ability to successfully meet complex demands in a particular context through the mobilisation of psychosocial prerequisites (**including cognitive and noncognitive aspects**)"—emphasis added [14]. Competencies capture the sense not only of acquiring but also of producing knowledge, embracing different ways of knowing and avoiding a narrow focus on specific skills [15]. We will suggest ways of interrogating the achievement of key competences and propose some subsequent steps to be taken within this iterative action-reflection process. We will also propose one route to operationalise our framework to monitor and evaluate ESD, using the multiple intelligences proposed by Gardner [16].

Specifically, we address the research questions: 'What competencies should education, in particular ESD, achieve to address SDG 4.7 and wider visions of sustainability?' and 'How do we begin to activate these competencies?'

## 2. Methods

We adopted a qualitative approach including iterations of dialogical analysis across our group. We drew on our different perspectives on sustainable development and on education in order to create a synthesis that was embedded in different forms of experience, pursuing the goal of becoming "critically reflective practitioners" [17]. 'We' were a community facilitator with experience of collective, value-based and practical action through deep interpersonal and intrapersonal engagement; a sustainability academic deeply connected with learning for sustainability through a UN Regional Centre of Expertise, integrating community, school, NGO, university and local government learning; a leading sustainability educator with experience in community, local authority and UN action; a philanthropist and economist specialised in alternative to international finance working in various branches of the business world; a senior European institutions civil servant engaged with community-led eco-development and good governance building; an academic researching eco-innovation and global futures; and an academic specialised in integral ecology, integrating cultural, socio-economic & policy dimensions in climate change adaptation research. The group thus brought together a unique suite of perspectives and experiential learning, with over 100 years of experience between them.

Our iterative dialogue included six meetings with between three and six participants, with some contributors meeting physically and others virtually using online platforms. Discussions varied in length from 30 min to 7 h. Additional communication was undertaken when finalising the writing of

this paper. The dialogues were structured around, firstly, the aims and scope of the work; secondly, a sharing of literature, concepts and practice informed contributions; thirdly, development and refinement of our framework; fourthly, testing of the framework concept with external stakeholders; and, finally, elaboration of our practical implementation and creation of the narrative to explain our process of analysis. In acknowledging the different contributions of practitioners and academics, we recognised the "complementary knowledges" and "ecosystem of expertise" required to pursue sustainability [18]. The iterative nature of the debate was inspired by the Delphi method [19] in that we addressed the same questions and then worked towards a consensus answer, nurtured through a consensus building approach [20]. Once we had reached a consensus framework, this framework was tested in an external workshop at the Bridge 47 conference in Brussels "Unlocking the Power of 4.7", 3 October 2018. The ESD concept and the evaluation framework were presented to 20 participants who were educators in both formal and informal education or representatives of NGOs. Participants were asked to critique the framework, accepting aspects with which they agreed and suggesting missing aspects. This led to validation of the framework and additional aspects for inclusion, as discussed in the results and discussion.

Below, we outline our framing of the aims and scope of the work through a brief discussion of the purposes of ESD, then define existing competency frameworks, articulate our suggested modifications, propose a mechanism for activation and outline an assessment framework.

## 3. Results

### 3.1. Purposes of Education for Sustainable Development

Whilst we began with the premise that competencies were effective for assessment of ESD outcomes, we started our dialogue questioning the purposes of ESD. Whilst first order learning is useful if stable and replicable knowledge transfer is needed, and the environment you are applying that knowledge or skill in is stable and unchanging, Sterling [21] proposes that second order learning is more appropriate for our changing world. By applying critical thinking skills (including normative and values analyses and systems thinking) the learner's worldview, values, and personal ways of knowing are challenged and changed accordingly. Sterling also draws on Bateson's [22] thinking around third order learning, which challenges the paradigm within which learning is taking place with a view to altering the learner's worldview. Sterling promotes this deeper, transformative learning, in which a shift of consciousness can occur and permit greater awareness not only of what and how to change the world, but why. If we acknowledge the need for transformative as well as first and second order learning, this enables us to consider what competencies might be required to imagine and practically pursue not only a change in the world, but also a change in oneself.

### 3.2. Key Competences for Sustainable Development

The idea of 'key competencies' has been proposed; competencies that are relevant across sectors and contexts [14] and that enable us to nurture 'change agents', 'problem solvers' and 'transition managers' [13]. Whilst competencies can be seen as "dispositions to self-organisation, comprising different psychosocial components", key competencies have special significance in that they have a wider focus across specific competency classes. Various combinations of key competencies have been proposed over the years. One suite drew on "Gestaltungskompetenz" developed in the late 1990s and comprised eight key competencies including future thinking, participatory abilities and capacity for empathy and compassion [23] (Table 1). Barth et al. [24] then proposed a suite of competencies including interdisciplinary working, transcultural understanding, capacity for empathy, compassion and solidarity and self-motivation (Table 1). Such key competencies are particularly valid if we see sustainable development as being a form of societal change that requires active participation by competent citizens; in this context, self development facilitates social advancement [24]. Podger et al. [25] referred to higher order dispositions rather than competencies and suggested

that ESD should educate the "whole person" and support sustainable "habits of mind", enabling dispositions such as systems thinking. Further work building on the key competencies of de Haan [23] and the 'dispositional thinking' of Reid et al. [26] offer classifications and syntheses of systematic, anticipatory, normative, strategic and interpersonal competencies [13] (Table 1). Wiek et al. [13] categorised competencies into clusters, which is a useful approach to enable contextualisation and development of specific competencies whilst ensuring categories are maintained. We draw on this latter synthesis for the skeleton of our framework, but we create a more vibrant living framework through our collective embodied experience as outlined below. Later competency frameworks or attempts at assessment have focused mainly on formal learning in higher education, with some on schools or teacher training, including those from Mochizuki and Fadeeva [27], UNECE [7] and Cebrian and Junyent [28], and in this issue Waltner et al. [29], Vare et al. [30] and Wilhelm et al. [31].

**Table 1.** Some of the suites of key competencies or outcomes that have previously been developed to assess Education for Sustainable Development (ESD) [13,24].

| Barth et al. 2007 | Weik et al. 2011 |
|---|---|
| **Key competencies in:**<br>Foresighted thinking<br>Interdisciplinary work<br>Cosmopolitan perception | **Anticipatory**<br>Developing narratives of the future<br>Backcasting and forecasting skills<br>Working with scenarios, risks, intergenerational equity, and unintended consequences |
| Transcultural understanding and cooperation<br><br>Participatory skills | **Systemic working**<br>Ability to work with key aspects of systems theory; tipping points, nested hierarchies and slow and fast variables and resilience |
| Planning and implementation<br>Capacity for empathy, compassion and solidarity<br>Self-motivation and in motivating others<br>Distanced reflection on individual and cultural models | **Interpersonal**<br>Including skills around mediation and conflict resolution<br>Leadership and team building<br>Communication skills, including empathy and empathic responses<br>Transcultural thinking and deliberation and negotiation |
| | **Normative**<br>The development of worldviews and perspectives<br>Ability to assess the stability of current or future states<br>Ethical questions, including risks and tradeoffs<br>Ability to assess well being |
| | **Strategic**<br>Planning, decision making, assessment of obstacles, identification of success factors<br>Knowledge of behavioural change<br>Organisational development<br>Use of Kolb's action reflection cycle. |

We can thus see that there is a range of perceived outcomes of ESD, varying from high level, broad, key competencies to specific indicators that enable us to rapidly assess what are often more superficial outcomes of ESD. Iterative dialogue amongst the co-authors raised questions. Firstly, whilst the academics realised the theoretical benefits of using competencies, others were concerned with pragmatic measurement aligning with existing audit efforts and thus wanted to focus on measurable indicators for the competencies with some meaning. Secondly, there was discussion of how the competencies might be used within non-formal learning contexts. Thirdly, there was debate over the different frameworks and exploration of 'missing values'.

These issues were explored in a series of four virtual workshops with community trainers involved with the Transition Towns network. A group of educators from six countries and all regions of the world except Africa and Antarctica met virtually to discuss essential skills for sustainability. They suggested that an area of human potential was not represented. These trainers felt that our original competencies omitted acknowledgement of the personal development and fulfilment that education should offer, including transformative potential. They thus proposed addition of intrapersonal competencies—our abilities to be aware of and be able to operationalise our inner landscape—in addition to the competencies suggested by Wiek et al. [13].

Participants discussed how ESD should facilitate an individual to equip themselves as a whole person and as a change agent, to effectively be able to function in a challenging world whilst protecting their core wellbeing, integrity and commitment. This notion derives from the sense of sustainable

development as an intention for societal change, and those educated in/for sustainable development as activists, or at least action oriented, and who pursue their vision for a sustainable world.

These additional intrapersonal competencies thus included recognition of the stressful, contradictory and even paradoxical situations in which individuals acted, as well as emphasising and developing the aspects of compassion and empathy as defined by de Haan et al. [23] and Barth et al. [24]. The workshop thus suggested the following intrapersonal competencies should be added:

- Presencing: The ability to stay present to your internal environment at the same time as engaging with your external environment.
- The ability to hold contradictory thoughts and feelings without having to resolve the contradictions.
- Knowledge of stress and how to know when you are stressed and what can help you to reduce your stress and avoid burnout.
- The ability to cultivate awareness; the skill to be present and out of that presence become aware of states of being beyond your rational mind.
- The knowledge and ability to find inner states of peace and compassion, for one's self and others.
- The ability to make meaning out of experience; and the ability to synthesise experience, models or frameworks, and feed back into previously unknown meta-perspectives.
- The ability to experience and deepen love and connection to yourself, other humans, and the non-human world.

### 3.3. Revised Key Competency Framework and Assessment

The initial suites of competencies were compared and analysed by coding forms of competencies. These are drawn mainly from the synthesis of Wiek et al. [13]. The intrapersonal competencies identified in our research were then added. We thus propose a summary set of competencies as illustrated below in Table 2. In addition, we developed example evaluation questions that could be used to bridge the gap between theoretical capacity and demonstrated measurable impact. Hence, for example, we give examples of competencies for the intrapersonal competency area, such as self-reflection, connection with self, mental wellbeing, and we illustrate what types of evaluation questions might enable educators or learners to reflect on the development of these competencies.

In this paper, we offer our evaluation framework as a set of questions (Table 2). Learners, practitioners, teachers and policy makers can respond to these questions in their own self-reflection and evaluation. This question format is important: It is open ended; it demands a thoughtful response and negates superficial 'box ticking'. Questions promote debate and dialogue and encourage honest and graduated responses. Table 2 demonstrates example questions that can probe the acquisition of key competencies, referencing multiple intelligences in how these are enacted. The questions as they are presented are framed around the education programme and presented for teachers, but they can be easily rephrased for individual assessment by learners. Hence, for example, "Are learners facilitated to work well with others?" might be rephrased as "Have you been supported to work well with others?" In addition, the questions in Table 2 are currently very broad, but they could be rephrased in relation to a particular programme or domain, e.g., engineering, lifestyle choices.

**Table 2.** Competencies proposed for assessment of ESD, including addressing Sustainable Development Goal (SDG) 4.7 (informed by Wiek et al. [13]) plus evaluation questions.

| Key Competency Area | Example of Competencies | Example Evaluation Questions |
|---|---|---|
| Intrapersonal | Presencing, self awareness, stress management, meaning making, connection with self, capacity for inner peace, mental wellbeing, self-reflection | Are learners able to be present in themselves? Can learners hold (without having to resolve them or prejudice one or the other) contradictory feelings and or thoughts? Do learners practise self awareness? Are learners able to know when they or a group is stressed and take appropriate steps so that stress does not hinder action? Can learners find strategies to seek inner peace? Can learners make meaning in the work they do? Do learners practise love and compassion? Are learners aware of their mental and emotional health and do they have the abilities to maintain healthy mental and emotional states? |
| Interpersonal | Communication skills, empathy, compassion, leadership, teamwork, mediation, cooperation, collaboration, participation | Are communication skills taught? Are learners facilitated to work well with others? Can learners assist each other in peer to peer learning? Are learners, across gender, ethnicity and other groupings, able to explore their leadership skills? Is empathy valued and encouraged? Are learners able to address conflict and develop mediation skills? Are their barriers to full participation in learning projects? |
| Future thinking | Visioning, developing scenarios, backcasting, recognising heritage, intergenerational equity | Are learners encouraged to imagine and envision sustainable futures? Can learners effectively use backcasting and forecasting skills in planning strategic activities? Do learners connect with their heritage and culture when looking to the future? Can learners identify future scenarios and use them to inform decision making? Are learners able to apply an awareness of intergenerational fairness to decisions and planning? |
| Systems thinking | Systems thinking, working with complex problems, promoting resilience, understanding tipping points and feedback loops | Are learners able to work with interconnectedness and complexity in a systemic context? Do learners have a functional knowledge of tipping points, resilience and feedback loops? Can learners understand how to work with socio-ecological systems? Do learners have a working concept of resilience? |
| Disciplinary and interdisciplinary | Understand the links between knowledge and experience, critical thinking, discipline specific framing, interdisciplinarity, expressing multiple ways of knowing | Have learners acquired an epistemological intelligence? Have learners developed awareness of different ways of knowing? Have learners explored disciplinary integrity and understood the academic norms of a discipline? Can learners work with disciplines that are not their core approach? Have learners developed their capacities for critical thinking? Can learners critically reflect on their own experiences? |
| Normative and cultural | Ethical responsibility, development of world views and perspectives, awareness of values, understanding of justice, cosmopolitan perception, transcultural understanding, awareness of local context and global trends | Can learners identify ethical questions and evaluate ethical responses according to different frameworks? Are fairness and justice debated and explored? Are learners encouraged to engage with and understand different world views? Are different cultural contexts appreciated? Have learners engaged with questions of well being and happiness? |
| Strategic | Planning, decision making, implementing, addressing challenges, organisational development, use of Kolb's action reflection cycle. | Are learners able to practise decision making and analyse consequences? Can learners use planning and assessment tools? Can learners identify and address challenges with regard to strategies and their implementation? Have learners implemented a plan they have designed? Do learners know how to use the behavioural change cycle for effective action and reflection? Are learners aware of organisational development issues and practices? |

### 3.4. Using the Competencies and Assessment Framework Against SDG 4.7

In further developing our evaluation framework in relation to the SDGs, as expressed as a goal of this paper, we now use the areas listed in SDG 4.7 and explore how we could assess our competencies against these, with cognisance of the multiple intelligences [16]. Whilst we recognise that fundamental competencies around communication and systems thinking are essential for ESD, we focus in this paper on developing a framework for competencies and examples of some of the specific skills required to address the five areas highlighted in the text of SDG 4.7 [2]:

- Human rights
- Gender equality
- Promotion of a culture of peace and non-violence
- Global citizenship
- Appreciation of cultural diversity and of culture's contribution to sustainable development

Example assessment questions can be seen in Table 3 with the competencies aligned with SDG 4.7 areas.

### 3.5. Activating Competencies through Multiple Intelligences

The competency frameworks suggested in this paper and elsewhere offer a route to assess ESD outputs. However, ESD is ultimately about facilitating transformative learning so that people can act sustainably. There are various possible approaches to enact ESD. We propose that the multiple intelligences concept offers a mechanism for learners to enact the competencies gained through ESD. The multiple intelligences proposed by Howard Gardner [16] represent a sum total of human capacities to influence, interact with, and communicate with our world. This includes both human and non-human life forms. The seven intelligences are represented in Figure 1.

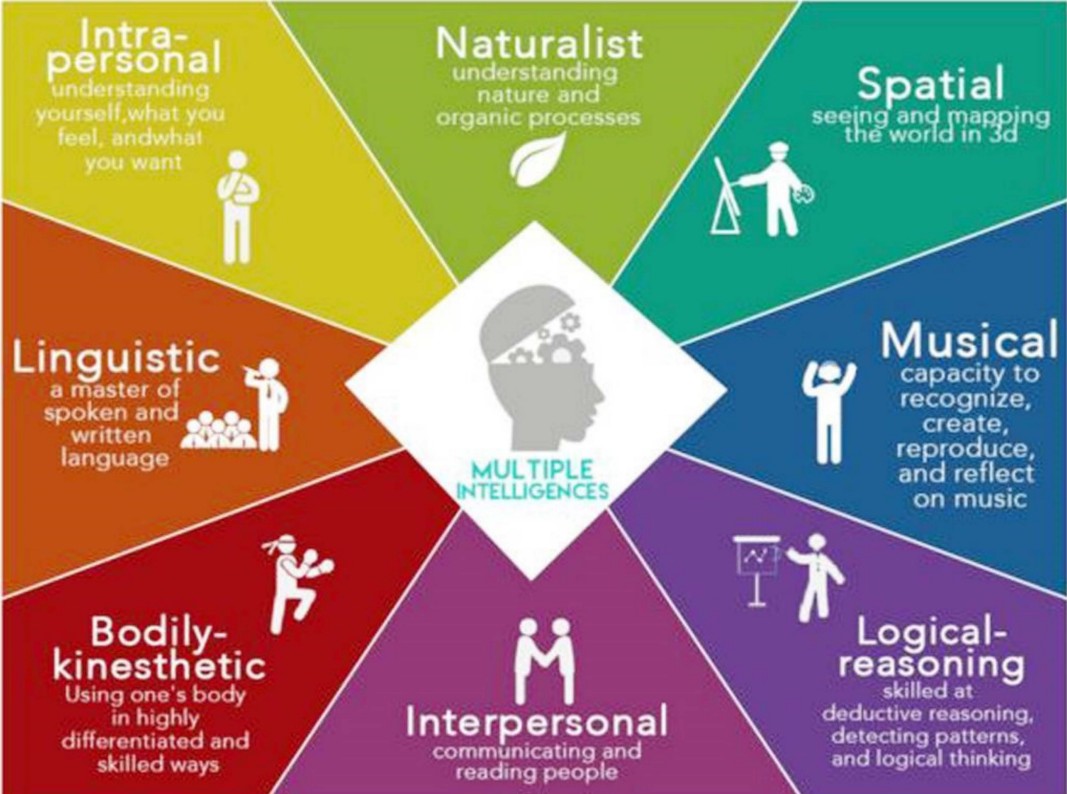

**Figure 1.** The multiple intelligence model developed by Gardner [16]—image http://thedailynewnation.com.

**Table 3.** Key competencies with reference to multiple intelligences, applied to the areas indicated in SDG4.7. These particular questions should be used together with the questions in Table 2.

| Key Competencies | Human Rights | Gender Equality | Culture of peace and non Violence | Global Citizenship | Cultural Diversity |
|---|---|---|---|---|---|
| **Interpersonal**. Collaborative skills, mediation, leadership, cooperation, empathy, teamwork | Are learners given the opportunity to develop empathy? What is leadership in promotion of human rights? What role does cooperation play in human rights? | Do learners have the possibility of experiencing their world from the viewpoint of the opposite sex? Do opposite sexes have opportunities to collaborate together on shared goals? Do both sexes look together at issues of gender equality in a spirit of enquiry? Are different forms of gender included and enabled? | Are there opportunities to explore peace or non-violence between groups or individuals in everyday contexts? Are any peace 'technologies' such as non-violent communication taught? | Can learners explore what it means to be a member of a particular social or ethnic group and a citizen of the world? How might that lead to differing actions or ways of thinking or being? Are learners given the opportunity to learn the international mechanisms of global cooperation? | Do learners have the opportunity to reflect and understand their culture's attitudes to others and 'otherness'? In what way can local cultural activities enable empathy and appreciation of cultural diversity? |
| **Strategic Planning**. Decision making strategies, awareness of success factors, obstacles to change, knowledge of behavioural change, organisational development skills | Do learners know what successful strategies for enjoyment of human rights have been employed in their countries? What is in the way of greater employment of human rights in their country? Can learners identify the changes in individual, group, or national behaviours that are most likely to lead to more human rights being enjoyed by more people? | Can learners find strategies that will lead to greater gender equality? What might be the obstacles to greater gender equality? Can learners identify what changes in organisations would lead to greater gender equality? | Can learners identify strategies for peace and non-violence? Can learners identify the barriers to peace at any levels of scale? Do learners experience differing organisational structures and what their role promoting peace and non violence might be like? | What strategies can lead to great engagement with global citizenship amongst their culture or country? What strategies might be employed to grow global citizenship awareness and action as opposed to nationalistic behaviour and attitudes? | Can learners express using cultural avenues ways to change behaviour or make change happen? |
| **Normative Competencies**. Knowledge of the sustainability of current or future states, knowledge of and awareness of Justice, fairness, happiness, well being, risk, trade offs, and ethical questions | Do learners explore where human rights come from and how? Can learners identify which human rights most directly affects personal happiness or well being? Do learners have the possibility of reflecting on the current level of human rights in their culture or country? Do learner reflect on the trade offs between different human rights and their consequences? | Can learners identify rightness or wrongness of the current state of gender equality? Do all genders have the possibility to fulfil their potential in education? Do learners identify In what way does gender determine levels of happiness or well being? | Can learners reflect on the relative sustainability of cultures of violence? Learners can reflect on how violence affects the sustainability of that society. Learners can reflect on the trade offs between violence and peace in their society. Learners can reflect on what risk a culture of peace carries. | Do learners have the opportunity to reflect on the risks and rewards of adopting global perspectives to themselves or their society? Can learners reflect on to what extent global citizenship is encouraged, or discouraged in their society? Do learners examine what improved global citizenship awareness might have in a an imagined future? | Can learners see the role their culture norms and values plays in promoting happiness, well being, justice, or fairness? Do learners reflect on how their culture engages with ethical questions and issues particularly around diversity? How are levels of diversity in their culture affecting general levels of well being? Can learners assess any risks or trade offs in their cultural diversity? |
| **Anticipatory Skills**. Working with scenarios, forecasting and backcasting, intergenerational equity | Can learners agree on a date in time where full implementation of Human rights be a reality and backcast the necessary steps until today, considering both incremental and transformative steps? Can learners imagine different scenarios or pathways of achieving plenum human rights in their country? | Have learners examined how inter generational equity has affected gender equality? | Can learners foresee a time in their country where a culture of peace and non-violence is a reality? Can they backcast these scenarios? Are they able to imagine or design different pathways with incremental or transformational steps? Do learners understand or know the process of intergenerational culture reproductions (memes) and ways of transforming them? | Can learners foresee a time when global citizenship has achieved equality with national citizenship | Can learners anticipate and outline pathways to a culture of equality of diversity? Can learners imagine or plot pathways to when questions of diversity become irrelevant? |

**Table 3.** *Cont.*

| Key Competencies | Human Rights | Gender Equality | Culture of peace and non Violence | Global Citizenship | Cultural Diversity |
|---|---|---|---|---|---|
| **System Thinking**. Ability to work with Feedback loops, systems and sub-systems, buffers and multiple variables, nested scales, resilience, and tipping points. | Do learners have opportunities to reflect on the role human rights play in changing systems of power or oppression? Can learners identify how human rights feature in human social or economic systems? | Are learners able to reflect on how social political and economic systems are distorted by gender inequality? What feedback loops might be operating in issues of gender in their culture? | Can learners build or model social or political systems of peace? What are the feedback loops (social, economic, political, or ecological) which create or maintain violence? Can learners reflect on levels of 'social capital' that need to be maintained to ensure a peaceful society? | How can system change knowledge be used to increase or decrease knowledge of or ability to act as a global citizen? | How do diversity issues influence tipping points in social, environmental, or cultural change? Is changing the diversity of a culture a key step towards initiating cultural change? |
| **Intrapersonal Competency**. Prescencing Ability to hold contradictory feelings and thoughts Personal and group stress management Cultivating awareness Finding inner peace and compassion, meaning making, Experiencing Love and connection | Can learners reflect on where the impulse for humans rights spring? What internal awareness or competencies enhance or detracts from the societal recognition of human rights? Can learners reflect on the effect human rights or the lack of human rights has on personal feelings of safety or peace? Can learners reflect on the role of love and compassion on human rights? | Can learners reflect on how the present level of gender equality affects their inner states of safety, compassion, stress, and connection? To what extent are cultural levels of stress affected by gender inequality? | Can learners reflect on peace and what levels of peace they find in themselves? Can learners reflect on where violence comes from in themselves and what makes a violent response more or less likely? Can learners hold or be present to violence with equanimity? Can learners reflect on the effect of a non-violent approach to communication has on their inner states? | Can learners reflect on what it feels like to them being both a person of a place and a global citizen? Can learners reflect on how an awareness of a global perspective changes their sense of themselves? Can learners reflect on how awareness global citizenship increases or decreases their levels safety or well being? | Can learners reflect on how their cultural values related to diversity increase or decrease their feels of presence? Can learners reflect on what effect cultural expressions like dance or singing has in their inner states? Can learners reflect on how cultural expressions like art or music can change their personal experience of being in a group? |

The multiple intelligences are expressions of human capacities. While they overlap the competencies to some extent (particularly in relation to intra-and inter-personal competencies), there is a difference between competencies, which are learned capacities, and the potentials (many of which are physically rooted) for interacting with our world, both internal and external, that multiple intelligences represent. Multiple intelligences are ways that we can operationalise a curriculum and enable key competencies to be expressed or realised in learners. Multiple intelligences recognise that learning is not only about cognitive but also non-cognitive aspects; education is not only about rational education. The multiple intelligence approach has been critiqued, e.g., [32], but we argue that diverse ways of physically, mentally and emotionally engaging with the world will facilitate learning and action.

## 4. Discussion

This research offers innovations to the concept of competencies to assess ESD, including the additional emphasis on intrapersonal competencies, the particular focus on community (non-formal) learning and on the SDGs, the nature of the assessment questions as part of an active learning cycle and the proposal for multiple intelligences to enact the competencies gained through ESD.

### 4.1. Intrapersonal Competencies Identified in Community ESD

The competency framework returns to the fundamental questions of what ESD is and why we need it. It then focuses on the outcomes in terms of the competencies gained by learners, rather than on programme specific information gains. Our framework builds on the work of previous researchers who have explored competencies [13,24] but it introduces a novel aspect in its articulation of the emphasis required on intrapersonal competencies. This innovation emerged through our dialogical and workshop piloting processes and because of the focus on community, non-formal learning. This suggestion aligns with the third order learning proposed by Sterling [21] and the transformation suggested by Bateson [22]. Bateson's L3, third order learning "challenges the interpretation of experience, relations, and truth systems, leading to broad questions such as human life, world ecology, and relations to higher powers." ESD in a transformative sense can challenge the learner's values and worldview—their 'self' at the most intimate level. ESD can provoke, and arguably requires, a shift in consciousness [21] which can be seen as questioning ourselves at such a deep level that new formations of self become apparent and integrated into our fundamental identity. Educating towards a critical moral consciousness permits the learner to acquire moral agency and can enable a spiritual awareness [25]. In this sense, it further contributes to thinking of how ESD can be transformative learning—it explicitly relates to the self and thus supports attributes and skills that facilitate self-reflection and transformation. This transformational potential emphasis could catalyse a step change in ESD. This framework acknowledges the importance of head, hands and heart proposed by Geddes [33]; whilst other ESD frameworks may focus more on the information gained (head) or hands on experience (hands). Certainly, we acknowledge the importance of these aspects of ESD in other competencies, but we propose that without the heart, the impetus to pursue sustainable development is lost and the capacity to do so is impeded. Our evaluation framework recognises the importance of our pursuit of SD and how our humanity itself is necessarily engaged in a cultural transformation.

Intrapersonal competencies can be a challenge for educators and learners. For example, student teachers in one study retained an impression of the primacy of green and environmental issues over ethical and emotional ones [28]. The non-cognitive competencies interact with cognition [24] and can be developed over time; as Vare et al. [30] point out, competence is a quality developed through practice and not an end state. Despite the challenge of naming and pursuing these less tangible outcomes, increasingly, revised competency frameworks appear to be highlighting aspects of interpersonal competencies. Some of these overlap: The "capacity for empathy, compassion and solidarity" [24], aspects of normative and collaboration competencies such as reflecting on one's values and empathetic

leadership [13], socio-emotional competencies [11], or a whole person approach including identity, motivation and higher order dispositions [25].

A further benefit of our framework is that it applies to all learning, whether formal (in schools, colleges, universities), non-formal (in communities, businesses) or informal (through media, cultural norms). The SDGs are universal, applying to all countries and all sectors. Education can and must underpin all of the goals, and ESD plays a major role in SDG 4.7. This framework thus supports the SDGs in totality. Community workers know that people need to achieve an emotional connection with sustainability issues to provoke action, and that personal change often results from sustainability action. In this research, community trainers identified the importance of intrapersonal competencies as evidencing sustainability learning. The non-formal learning context is important alone but also reinforces formal education [24].

Particular curricula and pedagogical strategies will still require more specific assessment [31] and it has been suggested that a combination of generic and situational competencies may offer a good framing [11]. However, we propose that this framework has potential for scaling, meaning it can also be used for national curricula and thus offers a potential basis for national progress monitoring against SDG 4.7. It can be used by policy makers, as well as teachers, community practitioners and learners themselves. National curricula and cultural contexts differ [34], and exploration of ontological, geographical and socio-economic differences will enhance our understanding of ESD and its relationship to the competencies. It will also support a more in depth debate between Global North and South over characteristics such as global citizenship [34].

### 4.2. ESD Assessment as Part of an Active Learning Cycle

The framework is not linear and is intended to be part of an iterative action/reflection cycle of planning, implementing, learning, reflection and re-planning. This framework recognises the positive learning cycle in monitoring and evaluation in which learning stimulates outcomes which are evaluated, feeding into reflection and planning for future implementation of teaching [35] (Figure 2). A novel feature of this framework is that it is designed for learners to self-assess, and is not merely for external evaluation by teachers and practitioners, although they can also use it. It thus strengthens this learning cycle and builds in second level learning, learning how to learn [21]. This notion of constant evaluative learning has been effectively used by Transition Network in their Transition Initiatives Healthcheck [36] to deepen impact and reach of initiatives as well as facilitating self-reflection. It is a virtuous cycle; self-reflection also builds intrapersonal competencies and enables learning across other competencies. Whilst such a self-reflective process can be more challenging for learners and requires skilled educators, it overcomes the problem of ESD competency assessment being perceived as a 'test' [29].

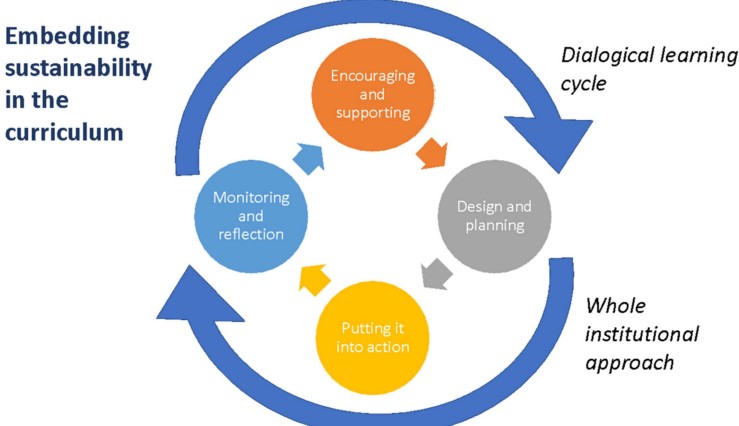

**Figure 2.** The action–reflection cycle of learning and assessment: Drawing on Kolb and Kolb [35].

### 4.3. Enacting Sustainable Development

We can thus see that our journey from discussion of the purposes of ESD to the desired outcomes and intentions of ESD to effective means of measurement has led us to a novel understanding. We align very closely with the frameworks for key competencies adopted by Wiek et al. [13], and Barth et al. [24] that draw on previous research in dispositional thinking [26] and competencies, but we also recognise the need for additional key competencies supporting intrapersonal transformation. However, if our expanded suite of key competencies defines a set of capacities that students can acquire and develop, how then do they activate this learning? The competences are generally framed as verbs, emphasising the need to do. In order to change competencies into action, we need a whole institutional approach, appropriate pedagogical approaches, specific methods and, importantly, for learners to understand their underlying values and motivational drivers. Recently, the utility of the terms "competence" and "capability" have actually been questioned because of ambiguity over whether a learner will have the motivation to enact newly acquired or polished competences [30]. Systemic critical thinking alone will not transform the world; moral motivation and action are thus required. Shephard et al. suggest that enacting learning may be more likely in specific contexts, for example after professional development; their study investigated health-care professionals. Individual centring is required to enact learning within societal orientation [24]. Throughout this paper we have articulated an understanding of ESD that accepts the need for societal change and for preparation of learners to make this change happen; there is thus a strong emphasis on action and even activism. We explored how these competencies might be enacted through Multiple Intelligences.

Enacting ESD knowledge and using competencies requires domain specific contextualisation. The domain specific (non-key) competencies should be already determined in each SDG area. For example, for SDG 14 the domain specific indicators represent the skills required as shown in Illustrative Organization of National, Regional, Global, and Thematic Indicators for SDG 14. It is critical that we acknowledge that the activation of competencies acquired in ESD occurs not only in specific domains, but also in different contexts and at different times within the life course. Hence, whilst ESD is important in schools, essential in colleges and universities, and invaluable in continued professional development, it is also required through non-formal and informal means of lifelong learning. 'Real life' learning also offers greater opportunities for experiential learning. Further, learning across domains and contexts offers new possibilities for systemic learning and action. For example, a whole institution approach at a university can be useful in linking teaching and learning, research, operations and community projects (such as Transition initiatives or Ecovillages) and wider external community engagement [37]. A further example is the knowledge exchange in pursuit of sustainability that occurs in Transition town initiatives and in activities such as skill-sharing [36].

This focus on the competencies and their enactment through multiple intelligences also emphasises the potentialities we expect of ESD and avoids prescriptive detailed instruction for either curriculum and pedagogy (which could thus impede creative, contextualised, place-based and disciplinary responses to ESD) or action (which should be context, time and domain dependent). However, our reluctance to dictate prescriptive means of imposing ESD is balanced by our recognition of the need for widely applicable evaluation frameworks of ESD. We thus propose a framework that is based on the competencies and intelligences derived above, and that can be used across a range of contexts and domains.

### 4.4. Challenges and Future Research for the Evaluation Framework

There are a number of challenges in developing and using this evaluation framework. Firstly, it largely produces qualitative outputs that can be difficult to reconcile with the quantitative indicators that nations and global institutions prefer to assess progress. There are mechanisms to quantify or codify text, but such approaches are often partially subjective. Secondly, evaluation takes time and effort, and there will be a tradeoff between specific knowledge gained and the emphasis on self-reflection and modification of the learning process. Thirdly, this evaluation framework demands

that individuals are honest with themselves and have capacity for self-reflection, traits that are not always favoured in contemporary formal education.

This evaluation framework is, of course, merely a framework. Future research should empirically test this framework to see how it functions in different practical situations, and we are planning this whilst hoping that others will also do so. We urge policy makers, in education but also with responsibilities for sustainability in general or the SDGs in particular, to test its efficacy. We encourage teachers to consider the framework when planning and assessing curricula. We suggest that practitioners with an interest in ESD (as widely defined) consider these aspects. Future research will also need to consider how these assessment questions can be left open yet at the same time be used to develop answers that can show enhancement of ESD over time. One mechanism for this may be to have answers given as yes/no, then with a self-assessment of extent, followed by a narrative of detail for each question.

Whilst these key competencies, including intrapersonal competencies, enable individuals to acquire capacities, abilities and skills to support sustainable development, their enactment will require additional assessment. Using a Multiple intelligences approach to curriculum development can ensure that the proposed key competencies can be enacted from ESD, but additional frameworks may need to be explored.

We also interrogated the question of how ESD can be integrated into the outputs and learnings from the other 17 SDGs, as well as how the evaluation framework can be continually assessed and those results integrated into a continually evolving evaluation framework. We are very far from a sustainable, let alone a regenerative world, and a process of continual assessment will be needed as we develop new practices, understandings and competencies for sustainable development in practice. This process of continual evolution is built into the SDG process (and the IPCC process) and it should also be instigated into this process of ESD. A representation of what this might look like is shown in Figure 3.

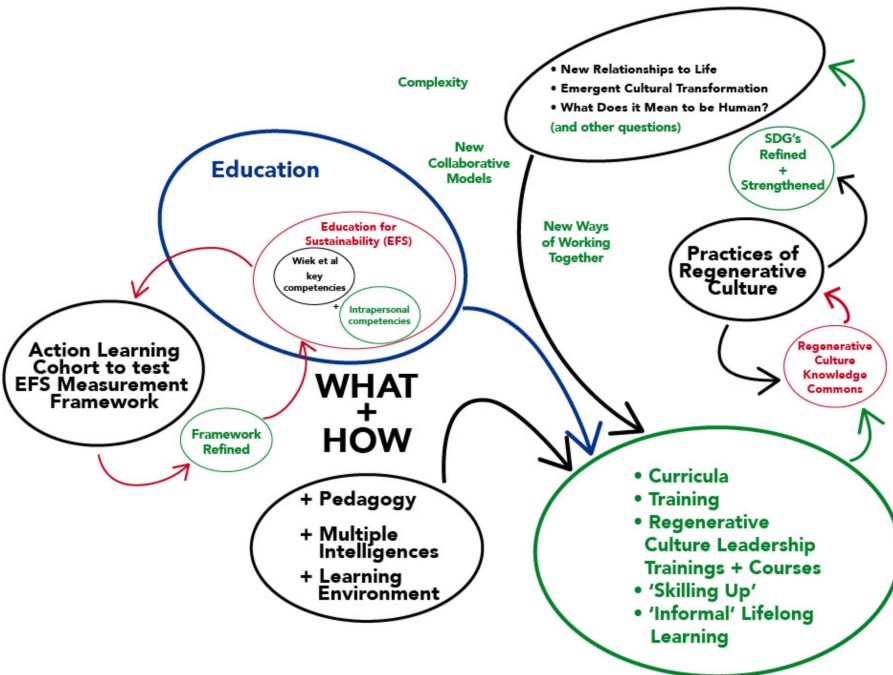

**Figure 3.** An illustration of how monitoring and assessment of ESD through competencies forms part of an active learning cycle, which together with pedagogy and learning environment integrate with curricula for formal and non-formal education. Education in this sense forms part of our wider practices for regenerative culture.

## 5. Conclusions

The pursuit of sustainable development is difficult, and it challenges us all at personal, professional and political levels. Few people would disagree that learning is fundamental to our journey, but there has been significant disagreement over how that learning might be delivered, experienced and assessed. Despite some healthy critique of the global agenda, the international community is now broadly coming together to support the UN SDGs. SDG 4.7 recognises the importance of ESD but does not offer clarity over what meaningful indicators might demonstrate that we are achieving this in different countries and contexts. We have put forward a competency framework that recognises that different forms of outcomes are desired. In particular, we emphasise the importance of intrapersonal transformation in enabling us to create the transformed world aspired to in Agenda 2030. We also suggest an assessment framework that emphasises an active learning aspect and propose enactment through application of multiple intelligences. Good evaluation schemes derive from solid goals, and the competencies allow us to define objectives for ESD and develop adaptive frameworks to support local and contextual learning whilst permitting the global analysis and monitoring required to address the SDGs. Perhaps paradoxically, in striving for global sustainability goals, we conclude that these can be achieved only through personal transformation and a shift in consciousness at an individual level, in which education must play an important role.

**Author Contributions:** Conceptualization M.E. and N.G.; methodology, R.M.W and N.G.; investigation and formal analysis, all authors; writing—original draft preparation, N.G. and R.M.W.; writing—review and editing, R.M.W., N.G., M.E., R.J., T.C., M.S.C., G.P.-L.

**Funding:** This research was supported by Gaia Education. G.P-L was funded by Fundação para a Ciência e Tecnologia (Contract number IF/00940/2015).

**Conflicts of Interest:** The authors declare no conflict of interest.

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
