# Peer review of "A Competency Framework to Assess and Activate Education for Sustainable Development: Addressing the UN Sustainable Development Goals 4.7 Challenge"

_sustainability, doi:10.3390/su11102832_

Reviewer 1 Report

I recommend that the purpose of the abstract be rewritten more focused on this study is too general.

In the methods section, recommending to mention the actual number of people who formed the study group

Lines 452-455. I recommend rewriting the title Fig. 1, is too long.

I recommend mentioning strength points and limitations.

It is inconsistent to number figures, please correct.

Author Response

please find our comments and revisions in the attached Pdf.

Reviewer 2 Report

Thaks for the paper. Some comments:

- in references, what is ln628 "85. eg. https://www.transitiontowntotnes.org/groups/skillshares/";

- be more precise in kaywors;

- I suggest a clarification of "We used a dialogical intervention across complementary expertises and piloted concepts in a trainer workshop. We developed a competency framework, drawing on previous models but innovating through the addition of intrapersonal competencies, a self-reflective validation scheme, a focus on non-formal learning and specific alignment with 4.7 requirements. Through exploration of how such learning could be activated, we proposed the use of multiple intelligences. It was concluded that, although the SDGs have a global remit, achieving the aspirations embedded within Agenda 2030 will require individuals to achieve personal  competencies, aligning with notions of transformational learning.";

- give a space in "[2].However,", and correct another typos like a extra space between reference 25 and 26;

- justify why: " We need a robust and deep reflective cycle with regards to ESD in general and Goal 4.7 in particular. . . . reflection and re-planning.";

- I don't understand why: "In this paper we emphasized noncognitive personal aspects of education and so we also drew on the definition that: “A competence is defined as the ability to  successfully meet complex demands in a particular context through the mobilisation of psychosocial prerequisites (including cognitive and noncognitive aspects)” - Emphasis added [21].";

- It seems limited the justifications of teh research questions (p. 3);

- There's an error in indication of "Figure 1";

- Clarify: "Figure 1: An illustration of how monitoring and assessment of ESD through competencies  f forms part of an active learning cycle, which together with pedagogy and learning environment integrate with curricula for formal and non-formal education. Education in this sense forms part of our wider practices for regenerative culture;

- Implications of this "Concept Paper"?

Author Response

please find our detailed comments and revisions in the attached table 

Reviewer 3 Report

1.good Job.

2.What does it stand for? ESD----page 2-----line(77).Make it more clear,please.

Perhaps author(s) can put ESD in keywords.(line 30 or 31).

3.Modify the citations or quotations.Some references are repeating.(for example Ref.42 and Ref

78)line---544 and line 614)

Suggested reference :1.An Empirical Study of How the Learning Attitudes of College Students toward English E-tutoring Websites Affect Site Sustainability. Sustainability 2019

Author Response

Please find our comments and revision sin the attached pdf

Reviewer 4 Report

This manuscript recognizes the difficulty in assessing Education for sustainable Development, and seeks to stablish a competency framework which allow not only assessment but also activating SDGs in practice.

The paper follows a qualitative approach based on dialogical interactions among actors representing very varied perspectives. After a consensus was reached, the authors propose a framework of competencies structured in 7 areas and with some research questions.

Authors are very right in proposing that ESD should be a goal for education and aim at transforming the society, and that this cannot be achieved just in formal educational contexts.  Overall, the paper is very relevant and well written, but there are some aspects I would like the authors to consider further.

Please see the attached document for more details.

Author Response

Please find our comments and revisions in the attached pdf

Reviewer 5 Report

An excellent idea but undeveloped enough in this stage of the manuscript. 

Even if the research questions are clearly formulated the methodology should be better and in details explained.

Also, after the introduction is expected a literature review part explaining the state of art in this area of research.

 The results part needs to be rewritten in a structured manner and focused on the results and the paper contribution in relation to the literature review section.

Related with Table 1 is needed a more deep discussion.

The information from Table 2 should be presented in a narrative manner

Table 3 is too large and should be moved into an appendix or split into components.

Author Response

Please find our detailed comments and revisions in the attached table.

Round  2

Reviewer 2 Report

In my opinion the paper now is sufficiente regurous and clear. The upgrade was evident.

Author Response

We have replied in this letter to all of your points.

Reviewer 5 Report

There are some improvements but still, the quality of the presentation is low and also the scientific soundness.

I expect some explanations at the previous recommendations that were just R ( I suppose rejected?) by the authors.

Author Response

We missed the replies to you points in the letter submitted, which the current letter includes. My apologies for that mistake. We have also replied to your round 2 questions, but we feel mostly they have been responded to in the corrected round 1 reply.
